# Insights into the Influence of Different Pre-Treatments on Physicochemical Properties of Nafion XL Membrane and Fuel Cell Performance

**DOI:** 10.3390/polym14163385

**Published:** 2022-08-18

**Authors:** Asmaa Selim, Gábor Pál Szijjártó, András Tompos

**Affiliations:** 1Institute of Materials and Environmental Chemistry, Excellence Centre of the Hungarian Academy of Sciences, Research Centre for Natural Sciences, Magyar Tudósok Körútja 2, 1117 Budapest, Hungary; 2Chemical Engineering and Pilot Plat Department, Engineering and Renewable Energy Research Institute, National Research Centre, 33 El Bohouth Street, Giza 12622, Egypt

**Keywords:** proton exchange membranes, Nafion XL, chemical treatment, hydration degree, dimensional swelling, water uptake, perfluorosulfonic acid/PFSA, fuel cell, chemical-physical properties

## Abstract

Perfluorosulfonic acid (PFSA) polymers such as Nafion are the most frequently used Proton Exchange Membrane (PEM) in PEM fuel cells. Nafion XL is one of the most recently developed membranes designed to enhance performance by employing a mechanically reinforced layer in the architecture and a chemical stabilizer. The influence of the water and acid pre-treatment process on the physicochemical properties of Nafion XL membrane and Membrane Electrode Assembly (MEA) was investigated. The obtained results indicate that the pre-treated membranes have higher water uptake and dimensional swelling ratios, i.e., higher hydrophilicity, while the untreated membrane demonstrated a higher ionic exchange capacity. Furthermore, the conductivity of the acid pre-treated Nafion XL membrane was ~ 9.7% higher compared to the untreated membrane. Additionally, the maximum power densities obtained at 80 °C using acid pre-treatment were ~ 0.8 and 0.93 W/cm^2^ for re-cast Nafion and Nafion XL, respectively. However, the maximum generated powers for untreated membranes at the same condition were 0.36 and 0.66 W/cm^2^ for re-cast Nafion and Nafion XL, respectively. The overall results indicated that the PEM’s pre-treatment process is essential to enhance performance.

## 1. Introduction

It is challenging to reliably and consistently apply the valuable electric energies provided by renewable sources such as solar and wind because of their instability and intermittency during generation. Using energy storage devices based on hydrogen and fuel cell technologies might significantly boost the utilization rate and stability of renewable energy sources, which would be a significant step toward solving this problem [1]. Proton Exchange Membrane Fuel Cells (PEMFCs) have made great strides toward becoming the next generation of green energy technologies [2,3,4].

Moreover, it is becoming more common for PEMFCs to be used in a variety of industrial applications, moving up to several hundreds of kW in stationary and transport systems from milliwatts in mobile devices [5,6,7]. Membrane Electrode Assemblies (MEAs) have three essential components; the porous gas diffusion layer responsible for the even distribution of reactants over the surface, electrocatalysts for the anode and a cathode side implementing the electrochemical reactions, and, between them there is the polymer electrolyte membrane or the proton exchange membrane (PEM). PEM acts as an electrolyte that transfers the protons from the anode to the cathode while preventing the conduction of electrons. The electrodes usually consists of carbon-based papers/felts and the electrocatalyst, mainly a Pt-based material [3,8,9,10]. In the operation, H_2_ and O_2_ or air is fed to the anode and cathode, respectively. H_2_ is oxidized over the anode catalyst producing protons (H^+^) and electrons (e^-^). PEM only allows the H^+^ to pass through the membrane while the electric current flows through an external electric circuit towards the cathode side. On the cathode side, water forms as a result of the reduction of O_2_ [11,12,13,14].

Hence, PEM has several functions during the process, such as separating the reactants to prevent the mixing of gases on both electrode sides, conducting the proton through it, and deflecting the electrons towards the external circuit. The proton exchange membrane is a critical component PEMFCs. The membrane has to have high ionic conductivity, low H_2_ crossover, and good thermal, electrochemical, and mechanical properties in dry and hydrated states [3,12,14,15,16]. Despite the advancements made at several levels in reducing its cost and improving its performance and durability, the need for ever-increasing longevity and lower costs necessitates the use of more robust components.

One of the most frequently used polymers for PEM fuel cells is perfluorosulfonic acid (PFSA) polymers. PFSA usually consists of the following two parts linked through an ether group: hydrophilic side chains with sulfonic acid groups and the hydrophobic backbone based on perfluorinated polytetrafluoroethylene (PTFE). The hydrophilic part is responsible for proton conduction in the hydrated state, whereas the morphological and mechanical stability are attributed to the hydrophobic backbone [17,18].

Nafion is the most studied and commercialized PFSA based polymer, and it was developed by DuPont de Nemours & Co in the 1960s. Nafion has high proton conductivity and acceptable chemical and mechanical stability at low and intermediate temperatures. The reason behind the high conductivity of Nafion is the existence of SO_3_H, which facilitates proton conduction through the hydrated membrane, as described before. In addition, the hydrophobic and hydrophilic domains which have been introduced result in significant phase separation/segregation between the two domains that form relatively wide micro-channels, which is beneficial to the transportation of protons [19]. However, as a result of the plasticizing effect of water on the Nafion chains, it has a significant hydrogen permeability in the hydrated state compared to the dry state [20,21]. Moreover, Nafion membranes are sensitive to both chemical and mechanical degradation. In the chemical process, hydroxide radicals formed as a byproduct of the electrochemical reaction attack the ionomer, resulting in its decomposition, whereas the swelling-induced mechanical stresses during hydration result in pinholes growth, delamination, and creep [22,23,24].

Recently, DuPont designed a membrane that shows advantageous mechanical-chemical stability called Nafion XL. Nafion XL is considered as a sandwich composite membrane consisting of two layers of Nafion impregnated with radical scavengers and, between them a microporous PTFE-rich support layer exists. The former layer can alleviate chemical attack through the radical scavengers, which can neutralize free radicals before they attack the ionomer. The latter is responsible not only for improving mechanical stability but also for reducing the dimensional swelling, which provides resistance to mechanical stress in the hydrated state. Nafion XL membranes were reported to have higher strength and toughness, less shrinkage stress, enhanced durability to creep and fatigue, as well as superior resistance during accelerated stress tests (AST) [23,24,25,26,27,28]. It is generally recommended in the related work to pre-treat such commercial membranes to improve performance [3]. The multi-step pre-treatment procedure usually consists of treatment with hydrogen peroxide in order to remove organic impurities, with deionized water used for rinsing, and protonation performed with an acid.

Before starting the fabrication of the MEA, the membrane was subjected to a treatment involving both water and acid. The goal of this study was to achieve a reference state for a commercial Nafion XL membrane. The outcomes were compared with those of a self-synthesized, “re-cast” Nafion membrane both before and after the pre-treatment, as well as those of an untreated Nafion XL membrane. The effect of the pre-treatment technique on the water uptake, dimensional swelling degree, ion exchange capacity (IEC), and hydration degree of the treated membranes was investigated. Conductivity and single fuel cell test results employing treated and untreated Nafion XL, as well as re-cast Nafion, are provided in order to verify the influence of the pre-treatment process on the electrochemical properties of the membranes. Results of re-cast Nafion tests were also reported.

## 2. Materials and Methods

### 2.1. Materials

DuPont Nafion solution (D520–1000 EW) containing 5 wt% copolymer resin was purchased from the fuel cell store. A commercial Nafion XL membrane with a thickness of ca. 27.5 μm was purchased from Ion Power GmbH, München, Germany. Dimethyl acetamide (DMAc) and sulfuric acid were obtained from VWR Chemicals, Budapest, Hungary. Carbon paper type H23C6 for the GDE preparation was obtained from Freudenberg FCCTSE&CO, Weinheim, Germany. Catalyst ink C-40-PT containing 40% Pt loading was purchased from QuinTech, Göppingen, Germany. Millipore water was used for all the membrane preparations was obtained in-house.

### 2.2. Membrane Preparation and Pre-Treatment

In this study, two PFSA membranes were used, namely commercial Nafion XL and laboratory prepared re-cast Nafion. For the re-cast Nafion, the solution casting process was used. Briefly, after evaporating the solvent from the Nafion solution, a suitable amount of the dissolved resin in DMAC was cast on a glass Petri dish. The produced membrane was dried at 80 °C for 24 h, followed by annealing at 120 for 4 h. The re-cast membrane was then obtained after immersing it in DI water for a few minutes. Both the re-cast and Nafion XL membrane were treated with deionized water and 0.5 M H_2_SO_4_. The pre-treatment process details are listed in Table 1. Considering the as-received and the re-cast membrane, deionized, acid-treated membranes are referred to as AsR, DI, and Acid throughout this paper, respectively.

### 2.3. Membrane Characterization

#### 2.3.1. Physicochemical Properties

Water uptake and the dimensional swelling ratio in terms of area (In-plane) and thickness (Through-plane) were calculated using Equations (1)–(3). Square membranes with a side length of 15 mm were used for these measurements, three measurements were performed at different locations and the average was obtained. First, the membrane samples were immersed in DI for 24 h at RT, then the surface water was removed with tissue paper and the wet mass and the wet dimensions of the membranes were noted. Finally, adsorbed water was totally evaporated by heating all the samples in a vacuum oven at 50 °C overnight, and the dry weight and size were recorded [29,30,31]. The water uptake (*WU*) was determined as follows:(1)$$WU\left(\%\right)=\left[\left(W_{w}-W_{d}\right)/W_{d}\right]\times 100$$
where $W_{w}$ and $W_{d}$ are the weights of wet and dry membrane samples, respectively. In-plane and through-plane swelling ratios were determined as follows:(2)$$In-plane\;SR\;\left(area\right)=\left[\left(A_{w}-A_{d}\right)/A_{d}\right]\times 100$$
(3)$$Through-plane\;SR\;\left(Thickness\right)=\left[\left(T_{w}-T_{d}\right)/T_{d}\right]\times 100$$
where $A_{w},A_{d},T_{w},T_{d}$ are the area and the thickness of the wet and dry membrane samples (cm), respectively.

Ion Exchange Capacity (IEC) was determined as the ratio of number of moles of sulfonic acid group per weight of dried membrane in gram. Acid-base titration was used to investigate the experimental values of IEC. First, all membranes were fully dried in an oven at 80 °C overnight, and weighed.

Secondly, the samples were soaked in 1 M NaCl solution for 24 h under continuous stirring after being cut into small pieces. Finally, the solutions were titrated with 0.01 M NaOH solution using methyl orange as an indicator. The IEC was calculated with the following equation [32]:(4)$$IEC=\left[\left(C_{NaOH}*V_{NaOH}\right)/W_{d}\right]$$
where $C_{NaOH}$ = 0.01 M, $V_{NaOH}$ is the volume of the NaOH solution used for titration and $W_{d}$ is the initial dry weight of the membrane.

Hydration degree, λ, is expressed as the number of water molecules available per SO_3_H group and considered as one of the essential characteristics for proton exchange membranes. The λ was determined for all the membranes from water uptake and ion exchange capacity from Equation (5) [33].
(5)λ=[(10∗WU)/(IEC∗18)]

#### 2.3.2. Electrochemical Characterization and MEA Performance

MEAs were prepared as follows. Gas Diffusion Electrodes (GDE) with 0.15 mg/cm^2^ Pt content were obtained by spray painting the catalyst ink onto the surface of the carbon paper; the catalyst ink consists of C-40-PT with 40 m/m% Pt content, Nafion solution (5 m/m%) and 2-propanol. For the negative electrode, anode, and for the positive electrode, cathode the Pt content was 0.15 mg/cm^2^. GDEs were heated at 80 °C before annealing at 120 °C for 30 min each. Membrane Electrode Assembly was prepared by pressing the membranes of 16 cm^2^ between two GDEs for 3 min under 59.4 kg/cm^2^ and 120 °C.

MEAs were characterized in single cells using VMP-300 multichannel potentiostat (BioLogic). Polarization curves were obtained after activating the membranes at 80 °C for 4 h at 0.4 V. The measurements were performed in a temperature range of 25–95 °C and under 50% and 30% relative humidity for H_2_ and O_2_, respectively. Back pressures were fixed at 250 kPa and 230 kPa for H_2_ and O_2_, respectively [34].

Membrane proton conductivity measurements were carried out at room temperature using potentiostatic electrochemical impedance spectroscopy (PEIS), and the frequency range was from 100 kHz to 10 mHz while the amplitude was 10 mV of oscillating voltage. Nitrogen and hydrogen flow was equal to 200 mL/min on both the cathode and the anode. Impedance spectra were recorded where the cathode was acting as the working electrode while the anode was considered as the unified reference and the counter electrode [35]. PEIS measurements were evaluated using the EC-lab program of BioLogic. Membrane resistance was calculated from the low intersect of the Nyquist plot with the z-axis. Membrane conductivity (S.cm^–1^) was obtained using the following equation:(6)σ=L/(R×A)
where L and A are the thickness and area of the membrane in cm and cm^2^, respectively, while *R* is the membrane resistance in Ω.

## 3. Results and Discussions

The thickness of the membrane was measured using a Mitutoyo micrometer and the dimensions of the membrane were measured with caliper feet.

Both re-cast and XL membranes had almost the same initial thickness of 27.8 and 27.5 μm, respectively. As shown in Figure 1, pre-treatment processes can result in a considerable increase in the thickness of the membranes. Specifically, the re-cast membrane shows approximately a 12 and 28% increase in its thickness after DI and Acid pre-treatment, respectively, whereas Nafion XL had a lower expansion in the thickness of approximately 5 and 15% for the same pre-treatments. The thickness increment may be due to the swelling of the ionic micelle nanostructure after the pre-treatment process. A similar trend was reported by Jiang et al. [36]. The smaller increase in thickness upon different treatments of Nafion XL may be due to the existence of the reinforcement layer and the radical scavengers impregnated into the Nafion XL structure hindering dimension change. Nevertheless, a moderate thickness growth may be beneficial for the chemical and physical properties of both the membranes, which is illustrated in the following chapter.

Generally, the performance of the polymer electrolyte membranes relies on its hydration and ability to retain water. The hydration properties of the membranes are usually investigated from the water uptake and dimensional swelling ratio and hydration degree. The influence of the pre-treatment processes on the swelling ratio both in-plane and through-plane, the water uptake of the membranes as well as the hydration degree results obtained in DI water at room temperature are presented in Figure 2a–c.

Based on the results reported in the literature, the different types of pre-treatments eventually result in higher water uptake in comparison to that obtained for the received samples due to the increase in the free volume available in the Nafion matrix. Upon pre-treatment, the membrane nanostructure was changed thereby allowing more water molecules to be absorbed [37]. The water uptake for the re-cast membranes is consistent with the previous reports [38]. The pre-treated membrane water uptake was very high compared to the AsR membrane (Figure 2a), and consequently the hydration degree was also higher (Figure 2c). Nevertheless, Nafion XL demonstrated only a slight increase in the water uptake (Figure 2a) as well as in the hydration degree (Figure 2c) after the pre-treatments. Most probably, the presence of the additives and the reinforcing layer affect the quasi-equilibrium state of the membrane during hydration.

On the other hand, to establish the influence of water uptake on the dimensional swelling of the membrane, the membrane swelling ratio in the in-plane and through-plane directions were obtained and the results are presented in Figure 2b. It is seen that both membranes possessed higher swelling ratios in both directions after pre-treatment. This is due to the fact that pre-treatment both with water or acid can erase the membrane thermal history and overwrite the hydrophilic passageways by connection previously isolated free volume voids via microscopic channels and allowing for the formation of large hydrophilic routes [39].

It is not surprising that Nafion XL exhibited larger expansion in thickness due to the presence of the reinforcement layer, which restricts the expansion in the in-plane direction. Additionally, the possibility of the anisotropic structure of the reinforcement layer itself in its plane can lead to an even greater through-plane swelling. Considering the fact that the in-plane swelling can result in cell failure due to the possible separation of the MEA components, Nafion XL has advantageous properties. However, the through-plane swelling can also lead to increased pressure between the MEA components, which can be treated by the extra compression of the MEA. Essentially, the through-plane swelling has a lower impact on the cell failure [40]. In general, from Figure 1 and Figure 2, the crucial role of the reinforcement layer and the additives in the XL structure can be concluded for the membrane stability even after the pre-treatment processes.

Ion Exchange Capacity is one of the most decisive factors influencing the resistance of an ion exchange membrane [36]. In Table 2, the IEC values for both re-cast and XL membranes before and after treatments can be seen. The values for a given membrane are in the same order of magnitude regardless of the different treatments applied. It is recognized that after DI treatment, the IEC values for both re-cast and XL membranes were somewhat lower than for untreated membranes, whereas the IEC values of acid-treated membranes were similar to that of AsR membranes due to the close pKa of the sulfuric acid and the sulfonic acid groups in the Nafion chain, which led to masking the effect of treatment [41]. The lower IEC values obtained after DI-treatment are a consequence of increased hydrophilicity and water uptake upon the pre-treatment process, which eventually can prevent the total acid capacity of the membrane being utilized, resulting in incomplete protonation and consequently lower IEC values [41,42,43].

The proton conductivity values of the as received and treated membranes at 25 °C were calculated from the impedance spectra and the Nyquist plot. The total resistance of the MEA was calculated from Equation (6) using the electrode resistance of 0.0228 Ω derived from the reported conductivity of the Nafion XL membrane by the supplier. Table 2 presents the obtained conductivity values for the treated and as-received re-cast and XL Nafion.

The conductivity of the PFSA membranes was reported to be improved by pre-treatment in water or acid. From the results, it can be seen that treating membranes in water and acid results in enhanced conductivity due to a greater water uptake, swelling ratio and hydration degree as discussed before. Eventually, a higher concentration of protons was obtained in the polymer matrix, which was most probably complemented by the formation of new conduction channels with different tortuosity and water networks. It is also recognized that the increase in the conductivity followed the same trend as the water uptake (Figure 2a).

Membrane electrode assemblies prepared from all treated and un-treated membranes were electrochemically characterized. The fuel cell performance was investigated for all the membranes using an in situ single fuel cell at the conventional temperature (80 °C) and relatively low humidity of 50% and 30% for H_2_ and O_2_, respectively. In Figure 3, the U-I polarization curves (a) and the power density curves (b) are plotted. Despite the lower IEC values, compared to the AsR membranes both water and acid treated re-cast and XL Nafion possess higher performance, which can be attributed to the higher affinity of the treated membranes to retain water leading to higher proton conductivity through the membranes.

Single cell efficiency was investigated by varying the operating temperature between 25 and 60 °C. Maximum power densities were obtained from the power density curves while the current densities were evaluated at 0.4 V from the polarization curves recorded according to the New European Driving Cycle protocol [43] and results are plotted in Figure 4.

As can be seen, the acid-treated membrane showed improved performance with increasing temperature. For all membranes, the highest power density was achieved at 80 °C. Additionally, the trends in current density change in Figure 4 (b) follow those for the maximum power density.

## 4. Conclusions

Commercial membrane Nafion XL and re-cast membrane (from Nafion solution) were treated in DI and sulfuric acid, followed by characterization and testing in a single fuel cell. Treatment in water is usually applied for cleaning membranes and improving their water affinity while treatment in sulfuric acid is usually used for enhancing the membrane hydrophilicity. Chemical-physical characterization was performed and the electrochemical properties were studied for all treated and untreated membranes. Acid-treated membranes showed a higher water uptake, in-plane, and through-plane swelling ratio compared to the as-received and prepared membranes. Nevertheless, the water uptake as well as the hydration degree of Nafion XL after the different pre-treatments was lower compared to those of the re-cast Nafion membrane as a result of the presence of a reinforcement layer in Nafion XL. The ion exchange capacity for the treated membranes was slightly lower than that for the as-received membranes, which however did not lead to the deterioration of proton conductivity. Finally, increased water uptake, degree of hydration, and the formation of new conduction channels resulted in better proton conductivity of the acid-treated membranes as compared to the as-received samples. As a result of the improved proton conductivity achieved after acid treatment, better performance of the commercial and the re-cast membranes was observed for the fuel cell tests compared to the results obtained on the as-received membranes.

## Figures and Tables

**Figure 1 polymers-14-03385-f001:**
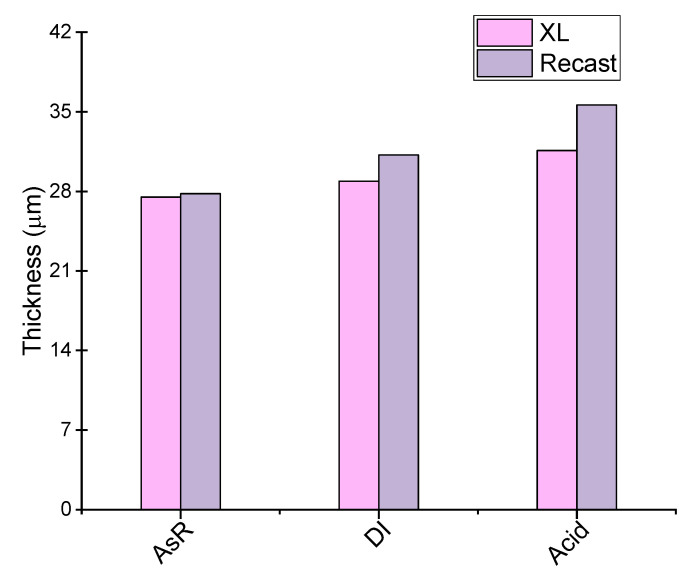
Influence of the different pre-treatments on the thickness of the re-cast and XL membrane.

**Figure 2 polymers-14-03385-f002:**
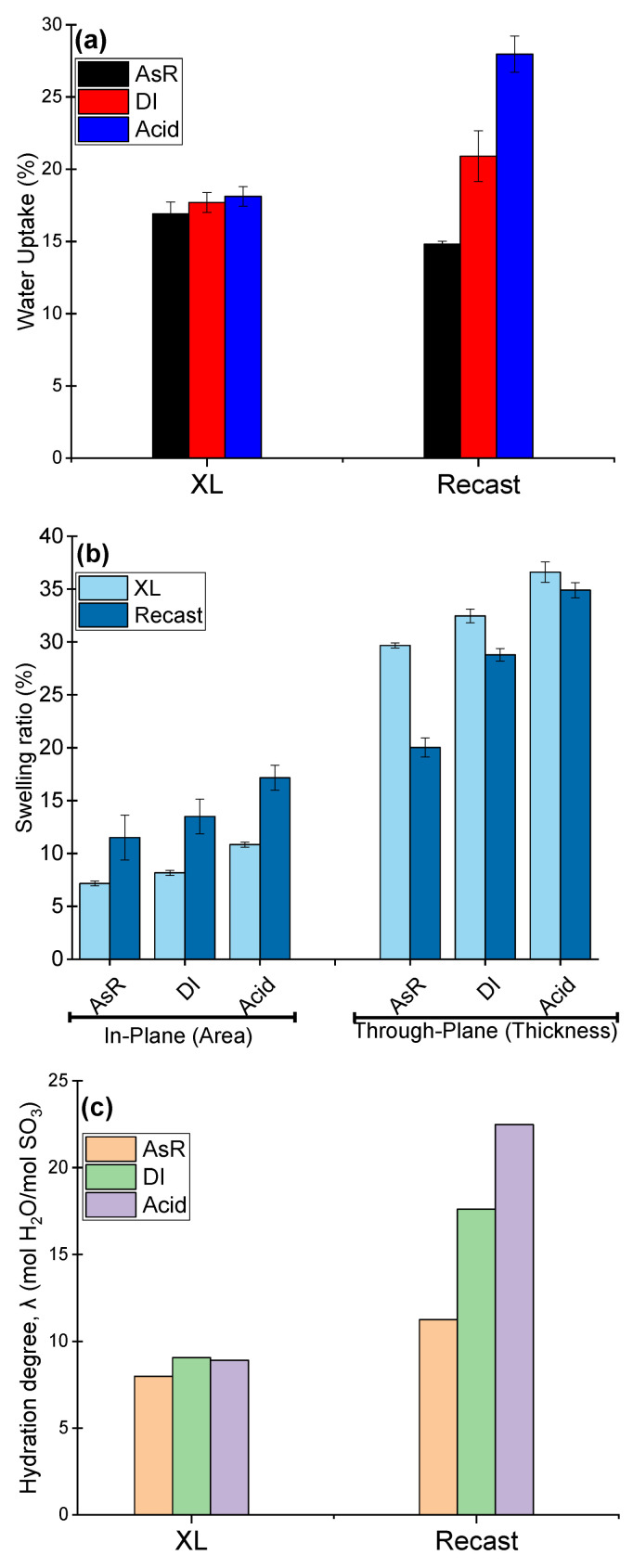
Hydration properties for re-cast Nafion and Nafion XL at different treatment, (**a**) Water uptake, (**b**) Dimensional swelling ratio, (**c**) Hydration degree.

**Figure 3 polymers-14-03385-f003:**
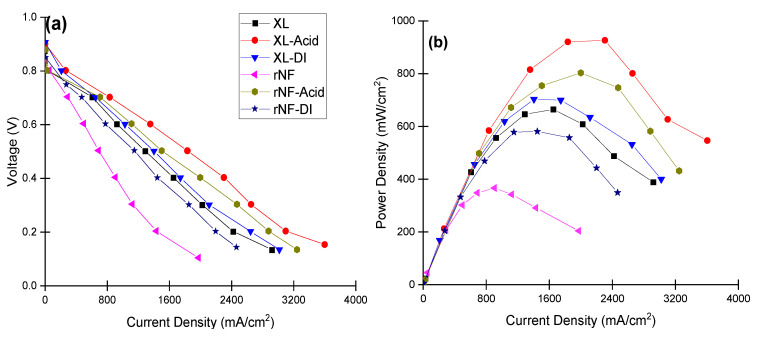
I-V (**a**) and power density (**b**) curves for all the membranes at 80 °C and relative humidity of 50%, 30% for H_2_ and O_2_, respectively.

**Figure 4 polymers-14-03385-f004:**
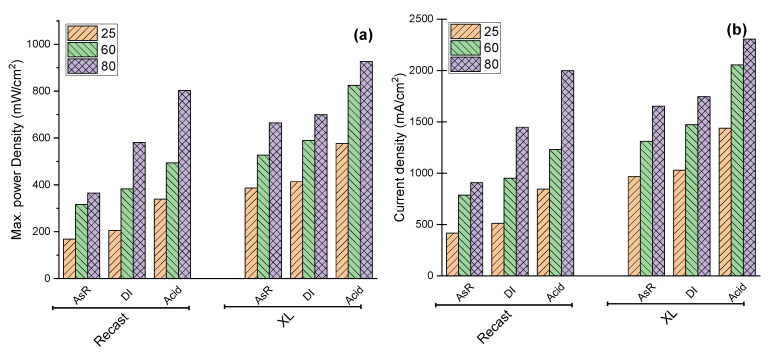
Influence of temperature on the single cell performance of treated and un-treated membranes at 25, 60 and 80 °C Maximum power density (**a**) and current density at 0.4 V (**b**).

**Table 1 polymers-14-03385-t001:** Nomenclature, a brief description of pre-treatment processes.

Nomenclature/Membrane	Pre-Treatment
AsR	Used as received or cast without any treatment
DI	Heated in DI water at 80 °C for 1 h
Acid	Heated in 0.5 M H_2_SO_4_ at 80 °C for 1 h

**Table 2 polymers-14-03385-t002:** IEC and conductivity values for untreated and treated re-cast and Nafion XL membranes.

Nomenclature/Membrane	IEC (mmol/g)		Conductivity (mS/cm)	
	Re-Cast	XL	Re-Cast	XL
AsR	0.73	1.18	4.89	50.55
DI	0.66	1.09	12.17	50.92
Acid	0.69	1.13	22.76	55.44

## Data Availability

The data presented in this study are available on request from the corresponding author.

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
