# Peer review of "Insights into the Influence of Different Pre-Treatments on Physicochemical Properties of Nafion XL Membrane and Fuel Cell Performance"

_polymers, 2022, doi:10.3390/polym14163385_

Round 1

Reviewer 1 Report

The manuscript reported the attempt designed to use a mechanical reinforced layer in the architecture and a chemical stabilizer to enhance the performance of Nafion, which acts as the most frequently used PFSA-based in PEMFCs. The influences of the water and acid pre-treatment process on chemical-physical properties of Nafion XL membrane and MEA were investigated. The conductivity of the acid pre-treated Nafion XL membrane was ~9.7% higher compared to the untreated membrane. The investigation of the maximum power density of the PEMFC at 80 °C was also carried out for comparison between treated and untreated Nafion membranes. The overall results elucidated that the pretreatment process of the PEM was essential to reach enhanced performance.

I consider the content of this manuscript will definitely meet the reading interests of the readers of the journal. However, the discussion and explanation should be further improved. I suggest giving a minor revision and the authors need to clarify some issues or supply some more experimental data to enrich the content. This could be comprehensive and meaningful work after revision.

1. For grammar issues, it is suggested that the author double-check the small grammar errors in the full text, especially the lack and redundant use of definite articles.

2. For the Keywords, ‘Perfluorosulfonic acid/PFSA’, ‘fuel cell’, and ‘chemical-physical properties’ should be added in order to attract a broader readership.

3. Line 29, ‘Proton Exchange Membrane Fuel Cells (PEMFCs) have made great strides toward 29 becoming the next generation of green energy technologies.’ I suggest that the connection between fuel cells and renewable energy be further emphasized here in the context of decarbonization and large-scale development and utilization of renewable energy nowadays.

Renewable energy sources such as solar energy and wind energy are unstable and intermittent during generation, and thus these valuable electric energies are difficult to apply continuously and stably. To tackle this issue, the employment of energy storage systems (such as PEMFCs) may greatly improve the utilization rate and stability of renewable energy [ChemSusChem 15.1 (2022): e202101798].

4. Line 43, ‘At cathode side, water forms as a result of reduction of O2. [10–13].Here it should also address the combination of the protons with the result of reduction of O2 (such as O2-) to form water. Otherwise, by only reduction of O2, water cannot be formed since where the H elements come from.

    In addition, ‘Hence PEM has several functions during the process such as separating the reactants to prevent mixing gases on both electrode sides, conducts the proton through it and deflects the electrons towards the external circuit.’ It should be made clear that the concept of selectivity, indicates that the proton side, should have as high as possible proton conductivity. While for the active species crossover side, the membrane should act as an efficient barrier layer to the hydrogen gas to prevent crossover as much as possible [Electrochimica Acta 378 (2021): 138133]. The higher the proton conductivity and the lower the hydrogen crossover will lead to a high membrane selectivity.

5. Line 59, ‘Nafion has high proton conductivity, …. The reason behind the high conductivity of Nafion is the existence of SO3H…This is not the only reason. Many polymers can also have SO3H groups, but the proton conductivity cannot be as high as Nafion, such as SPEEK (Degree of Sulfonation <60%, otherwise swelling easily). Since the hydrophobic and hydrophilic domains have been introduced, it should also refer to the significant phase separation/segregation between the two domains that form relatively wide micro-channels, which is beneficial to the transportation of protons [Solid State Ionics 319 (2018): 110-116]. That is why Nafion has so high proton conductivity.

6. In this paper, why is only recast Nafion compared with Nafion XL, not also with commercial pristine Nafion membranes? The commercial Nafion membrane is usually prepared by extrusion, which is obviously different from the solution casting method. Is the data of Nafion prepared by solution casting method and commercial sandwich-type Nafion XL prepared by extrusion method really comparable? I think this issue should be explained clearly in the manuscript, or the author needs to supplement the data on commercial Nafion membrane.

7. Why DMAc is selected as the only solvent for solution casting? This should be explained better to convince the readers since it has been reported that ‘Nafion membrane with DMF solvent has the best characterization and performance’ [International journal of hydrogen energy 41.1 (2016): 476-482].

8. In Part 2.2, it seems that Nafion pretreatment mode is relatively few and monotonous [see Table 1 of ACS applied materials & interfaces 8.19 (2016): 12228-12238], and it is almost not mentioned whether H2O2 is used in pretreatment. These issues need to be clarified.

9. Line 131, ‘Secondly, the samples were soaked in 1 M NaOH solution for 24 h under continuous stirring after being cut into small pieces. Finally, the solutions were titrated with 0.01 M NaOH solution using methyl orange as an indicator.I suggest double-checking this process. It should be soaked in 1M NaCl, and later titration with 0.01M NaOH. It does not make any sense if both are NaOH. Imagine that if the Nafion is soaked in NaOH at the beginning, so the protons will be reacted completely with OH- to form H2O. And for the latter titration with 0.01M NaOH, where is the remaining H+ to react with OH-?

Author Response

  1. For grammar issues, it is suggested that the author double-check the small grammar errors in the full text, especially the lack and redundant use of definite articles.

Thank you for this suggestion. The English of the manuscript has been improved by an external proofreading service.

  1. For the Keywords, ‘Perfluorosulfonic acid/PFSA’, ‘fuel cell’, and ‘chemical-physical properties’ should be added in order to attract a broader readership.

Thank you for this suggestion. The suggested keywords have been added.

  1. Line 29, ‘Proton Exchange Membrane Fuel Cells (PEMFCs) have made great strides toward 29 becoming the next generation of green energy technologies.’ I suggest that the connection between fuel cells and renewable energy be further emphasized here in the context of decarbonization and large-scale development and utilization of renewable energy nowadays.

Renewable energy sources such as solar energy and wind energy are unstable and intermittent during generation, and thus these valuable electric energies are difficult to apply continuously and stably. To tackle this issue, the employment of energy storage systems (such as PEMFCs) may greatly improve the utilization rate and stability of renewable energy [ChemSusChem 15.1 (2022): e202101798].

Thank you for this valuable comment and information. The appropriate text has been added to the manuscript.

  1. Line 43, ‘At cathode side, water forms as a result of reduction of O2. [10–13].’ Here it should also address the combination of the protons with the result of reduction of O2(such as O2-) to form water. Otherwise, by only reduction of O2, water cannot be formed since where the H elements come from.

Thank you for this valuable comment. The combination with proton is already stated in lines 41-42.

“H2 is oxidized over the anode catalyst producing protons (H+) and electrons (e-). PEM allows only the H+ to pass through the membrane while the electric current flows through external electric circuit towards the cathode side.”

    In addition, ‘Hence PEM has several functions during the process such as separating the reactants to prevent mixing gases on both electrode sides, conducts the proton through it and deflects the electrons towards the external circuit.’ It should be made clear that the concept of selectivity, indicates that the proton side, should have as high as possible proton conductivity. While for the active species crossover side, the membrane should act as an efficient barrier layer to the hydrogen gas to prevent crossover as much as possible [Electrochimica Acta 378 (2021): 138133]. The higher the proton conductivity and the lower the hydrogen crossover will lead to a high membrane selectivity.

Thank you for this suggestion. The selectivity of the membrane in terms of high conductivity and low crossover of hydrogen gas is already stated after the mentioned sentence

“The membrane has to have high ionic conductivity, low H2 crossover, good thermal, electrochemical, and mechanical properties in dry and hydrated states [3,12,14–16]”

  1. Line 59, ‘Nafion has high proton conductivity, …. The reason behind the high conductivity of Nafion is the existence of SO3H…’ This is not the only reason. Many polymers can also have SO3H groups, but the proton conductivity cannot be as high as Nafion, such as SPEEK (Degree of Sulfonation <60%, otherwise swelling easily). Since the hydrophobic and hydrophilic domains have been introduced, it should also refer to the significant phase separation/segregation between the two domains that form relatively wide micro-channels, which is beneficial to the transportation of protons [Solid State Ionics 319 (2018): 110-116]. That is why Nafion has so high proton conductivity.

Thank you for this valuable comment and information. The appropriate text has been added to the manuscript.

  1. In this paper, why is only recast Nafion compared with Nafion XL, not also with commercial pristine Nafion membranes? The commercial Nafion membrane is usually prepared by extrusion, which is obviously different from the solution casting method. Is the data of Nafion prepared by solution casting method and commercial sandwich-type Nafion XL prepared by extrusion method really comparable? I think this issue should be explained clearly in the manuscript, or the author needs to supplement the data on commercial Nafion membrane.

Thank your for grapping the attention to this point. The aim of this work was to study the effect of the pretreatment processes on recast membrane (Homemade) and comparing with the available commercial membrane. In addition to the effect of these pretreatments on the fuel cell performance. But of course the way of the membrane preparation could have an effect on the results. However, authors believe that the effect of the manufacturing method would not have such a large impact on the fuel cell performance.

  1. Why DMAc is selected as the only solvent for solution castingThis should be explained better to convince the readerssince it has been reported that ‘Nafion membrane with DMF solvent has the best characterization and performance’ [International journal of hydrogen energy 41.1 (2016): 476-482].

Thank you for this comment. DMAc also was applied as good solvent for Nafion, DMAc has been known as a good dispersion solvent for unsupported Pt catalysts and that it improves the proton conductivity of the cast membrane by interacting with the polymer .

According to the mentioned article, authors were changing solvent (DMF, DMAc and NMP) for Nafion and the solvent effect on the Nafion polymer containing TiO2 as a nanofiller, however, from some point of views DMF has more advantages than DMAc, but that does not rule out that DMAc is still good solvent for Nafion.

  1. In Part 2.2, it seems that Nafion pretreatment mode is relatively few and monotonous[see Table 1 of ACS applied materials & interfaces 8.19 (2016): 12228-12238], and it is almost not mentioned whether H2O2 is used in pretreatment. These issues need to be clarified.

Thank you for this comment.

The mentioned paper is an excellent reference for various pretreatments and has been used as reference in this manuscript. However, the pretreatment in this manuscript is mainly dealing with water treatment to hydrate the membrane, and the protonation step in which the membranes are boiled in H2SO4. So the H2O2 is not used as pretreatment in this work. However, the authors will consider this valuable comment.

  1. Line 131, ‘Secondly, the samples were soaked in 1 M NaOH solution for 24 h under continuous stirring after being cut into small pieces. Finally, the solutions were titrated with 0.01 M NaOH solution using methyl orange as an indicator.’ I suggest double-checking this process. It should be soaked in 1M NaCl, and later titration with 0.01M NaOH. It does not make any sense if both are NaOH.Imagine that if the Nafion is soaked in NaOH at the beginning, so the protons will be reacted completely with OH- to form H2O. And for the latter titration with 0.01M NaOH, where is the remaining H+ to react with OH-?

Thank you for this comment. Actually, this is just a typo and was corrected in the manuscript.

Reviewer 2 Report

Dear Authors,

You did great work, but the article must be considerably improved.

The introduction of the paper named “Insights into the Influence of different pretreatment on chemical-physical properties of Nafion XL membrane and fuel cell 3 performance” is well argued but still should be improved. Extensive English corrections must be made to be published.

Starting with the title, I consider that authors should also change “chemical-physical” into “physicochemical” within the manuscript.

The paper lacks characterization techniques, and applications are not very well illustrated. Maybe a paragraph should be about what applications are used for these membranes.

The first part of the results presents a figure regarding the thickness of these membranes. How was this thickness measured?

Also, other techniques for characterization are needed, such as XRD, TGA, FTIR, SEM or NMR…Pore size and dimensions are essential measurements in such a study.

 After reading and comprehending the information presented in the paper, I consider it subjected to major revisions and considerably improved before publication.

Author Response

1-The introduction of the paper named “Insights into the Influence of different pretreatment on chemical-physical properties of Nafion XL membrane and fuel cell 3 performance” is well argued but still should be improved. Extensive English corrections must be made to be published.

Thank you for this suggestion. The English of the manuscript has been improved by an external proofreading service.

2-Starting with the title, I consider that authors should also change “chemical-physical” into “physicochemical” within the manuscript.

Thank you for this suggestion. The title has been changed.

3-The paper lacks characterization techniques, and applications are not very well illustrated. Maybe a paragraph should be about what applications are used for these membranes.

Thank you for this suggestion. The last paragraph in the introduction section has been modified to emphasize the application of these membranes.

4-The first part of the results presents a figure regarding the thickness of these membranes. How was this thickness measured?

Thank you for this question. The thickness of the membrane was measured by Mitutoyo digital micrometer and the dimensions of the membrane were measured by caliper feet.

This information has been added to the manuscript.

5-Also, other techniques for characterization are needed, such as XRD, TGA, FTIR, SEM or NMR…Pore size and dimensions are essential measurements in such a study.

Thank you for this comments, most of these characterization techniques for both Nafion XL and recast Nafion membrane have been studied and published in our previous published paper in the same journal.

So, in this manuscript authors tried to focus more on influence of pretreatment on the dimensional swelling, physicochemical properties such as IEC, and hydration degree ,electrochemical properties such as conductivity, and fuel cell performance.

Round 2

Reviewer 2 Report

Dear Authors,

I want to thank you for taking the time to answer my suggestions.

I think the manuscript fulfils all the requirements for publishing.

Keep up the excellent work!